# Controllable Synthesis of N_2_-Intercalated WO_3_ Nanorod Photoanode Harvesting a Wide Range of Visible Light for Photoelectrochemical Water Oxidation

**DOI:** 10.3390/molecules28072987

**Published:** 2023-03-27

**Authors:** Dong Li, Boyang Lan, Hongfang Shen, Caiyun Gao, Siyu Tian, Fei Han, Zhanlin Chen

**Affiliations:** 1School of Material Science and Engineering, North Minzu University, Yinchuan 750021, China; 2International Scientific & Technological Cooperation Base of Industrial Waste Recycling and Advanced Materials, Yinchuan 750021, China; 3Chemical Science and Engineering College, North Minzu University, Yinchuan 750021, China

**Keywords:** N_2_-Intercalated, water oxidation, tungsten trioxide, photoelectrochemical, water splitting

## Abstract

A highly efficient visible-light-driven photoanode, N_2_-intercalated tungsten trioxide (WO_3_) nanorod, has been controllably synthesized by using the dual role of hydrazine (N_2_H_4_), which functioned simultaneously as a structure directing agent and as a nitrogen source for N_2_ intercalation. The SEM results indicated that the controllable formation of WO_3_ nanorod by changing the amount of N_2_H_4_. The *β* values of lattice parameters of the monoclinic phase and the lattice volume changed significantly with the n_W_: n_N2H4_ ratio. This is consistent with the addition of N_2_H_4_ dependence of the N content, clarifying the intercalation of N_2_ in the WO_3_ lattice. The UV-visible diffuse reflectance spectra (DRS) of N_2_-intercalated exhibited a significant redshift in the absorption edge with new shoulders appearing at 470–600 nm, which became more intense as the n_W_:n_N2H4_ ratio increased from 1:1.2 and then decreased up to 1:5 through the maximum at 1:2.5. This addition of N_2_H_4_ dependence is consistent with the case of the N contents. This suggests that N_2_ intercalating into the WO_3_ lattice is responsible for the considerable red shift in the absorption edge, with a new shoulder appearing at 470−600 nm owing to formation of an intra-bandgap above the VB edges and a dopant energy level below the CB of WO_3_. The N_2_ intercalated WO_3_ photoanode generated a photoanodic current under visible light irradiation below 530 nm due to the photoelectrochemical (PEC) water oxidation, compared with pure WO_3_ doing so below 470 nm. The high incident photon-to-current conversion efficiency (IPCE) of the WO_3_-2.5 photoanode is due to efficient electron transport through the WO_3_ nanorod film.

## 1. Introduction

In order to solve both the energy crisis and environmental problems, it is urgent to explore a sustainable, and environmental-friendly energy resources to replace the non-renewable fossil energy [1,2,3]. Hydrogen, as an energy carrier of solar energy, has the advantages of high energy density, storage, and pollution-free. So, it can be used as an ideal new energy to replace the traditional energy. To date, among the different strategies for producing hydrogen, PEC water splitting is considered to be a promising process in which solar energy can be directly convert into hydrogen and oxygen using a semiconductor [4]. However, the overall efficiency of PEC water splitting is still relatively low due to the high kinetic overpotentials for the water oxidation reaction. Therefore, the key to improving the efficiency of PEC water splitting is to develop a robust semiconductor photoanode for visible-light-driven water oxidation. Since TiO_2_ was first reported as photoanode by Fujishima and Honda in 1972 [5], numerous semiconductors (WO_3_, Fe_2_O_3_, Ta_3_N_5_, TaON, etc.) have been employed for photoanode materials [6,7,8,9,10,11,12,13].

WO_3_ was investigated as a visible-light-driven photoanode material for PEC water oxidation by Hodes in1976 [14]. Since then, considerable efforts [15,16,17,18,19,20,21] have been devoted to investigating the PEC performances of WO_3_ due to the following intrinsic properties: (1) a relatively narrow band gap energy (2.5 eV–2.7 eV), (2) a positive valence band edge position for water oxidation, (3) high stability in acidic condition, (4) a suitable hole diffusion length (~150 nm) for superior electron transport. However, the solar spectrum utilization of WO_3_ photoanode is still too low to be used for actual application due to its narrow visible light response range (λ < 460 nm). To enhance the PEC performance, many tactics have been employed to develop efficient WO_3_ photoanode for water oxidation, including forming heterojunction, doping, loading co-catalysts, nanostructuring, tuning vacancies and so on [1,4,22,23,24,25,26,27,28,29,30]. Nanostructured WO_3_ photoanodes have attracted more and more research attention to improve their PEC performance, because they offer superior performances than that of unspecified ones because of larger electrode/electrolyte interface area, efficient light absorption, and other structural benefits of nanostructures. So far, various nanostructured WO_3_ photoanodes have been synthesized, such as nanosheet [31], nanoplatelets [32], nanoparticles [33,34], nanoflakes [35,36], nanotubes [37], nanobelts [38], nanowire [39,40], and nanorods [41,42], to improve performances in PEC water oxidation. Although the PEC performance of WO_3_ photoanode for water oxidation can be enhanced by nanostructure control, the light absorption at long wavelength is difficult to improve. Nowadays, many studies are focusing on extension of light absorption to longer wavelength (*λ* > 480 nm) by band gap engineering of WO_3_, because efficient light absorption at longer wavelength is significant to improve the efficiency of a photoanode for solar energy conversion. To date, doping WO_3_ with transition/other metals [43,44,45,46], nonmetallic elements [25,26,47], or selective molecule [27] to improve its the light absorption at longer wavelengths have been regarded as the most common strategies for band gap engineering.

We previously reported an in situ N_2_-intercalated WO_3_ nanorod photoanode, in which N_2_H_4_ was used as a dual-functional structure-directing agent for nanorod as well as a nitrogen source for in situ N_2_ intercalation [17]. Not only the light absorption at longer wavelengths, but also the PEC performance for water oxidation was dramatically improved. Therefore, a difficult issue of compatibility between nanostructure control and strategy for band gap engineering of WO_3_ was resolved. However, we only discussed the calcination temperature dependence on the physiochemical properties. The PEC performance, the effect of addition of N_2_H_4_ on the content of N_2_ into the WO_3_ lattice, as well as the morphology of the N_2_-intercalated WO_3_ and the PEC performance for water oxidation have not been investigated yet. Thus, it is necessary to reveal the effect of addition of N_2_H_4_ on the morphology and band gap of N_2_-intercalated WO_3_. Herein, we first report the controllable synthesis of N_2_-intercalated WO_3_ with different morphologies and band gap using N_2_H_4_ as a dual-functional surfactant template. Especially, there is an extremely significant mutually dependent relation between the addition of N_2_H_4_ and the PEC performance of N_2_-intercalated WO_3_ for water oxidation.

## 2. Results and Discussion

### 2.1. Characterization Structure of N_2_-Intercalated WO_3_ Samples

The SEM images showed different morphologies depending on the addition of N_2_H_4_. WO_3_-0 is composed of small irregular nanoblocks with diameters of 10−50 nm (Figure 1a). Figure 1b−f shows the images of N_2_-intercalated WO_3_ prepared using different addition of N_2_H_4_. It can be clearly found that the mixed morphologies of nanorods in large scale and small nanoblocks start to be observed from n_W_:n_N2H4_ of 1:0.62 (Figure 1b). There are not so many differences in the length and width of the nanorods which are obtained from n_W_:n_N2H4_ ratio of 1:1.25 to 1:2.5. However, the length and width increased as the n_W_:n_N2H4_ ratio increased from 1:5 to 1:7.5.

The specific surface area of WO_3_−2.5 was 2.2 times higher than that of WO_3_-0 (9.6 m^2^g^−1^). The surface area of N_2_-intercalated WO_3_ samples decreased to 20.4–17.3 m^2^g^−1^ with increasing n_W_:n_N2H4_ ratio from 1:5 to 1:7.5 due to the agglomeration of nanorod surfaces.

The EDS data of N_2_-intercalated WO_3_ indicate the existence of the N element, which derived from N_2_H_4_ in precursors, because none of the N element was detected in WO_3_-0; see Figure 2.

The W/N atomic ratio was calculated from the EDS data to reveal the dependence of the n_W_:n_N2H4_ ratio on it. The results indicate that the W/N atomic ratio increased from 1:0.004 to 1:0.096 with increasing n_W_:n_N2H4_ ratio from 1:0.62 to 1:2.5 and thereafter decreased over the 1:2.5 (Table 1).

To instigate the effect of the addition of N_2_H_4_ on the crystal structures, the N_2_-intercalated WO_3_ samples are ascertained by XRD (Figure 3A) and Raman (Figure 3B). As shown in Figure 2A, it can be clearly observed that all of samples exhibited the monoclinic WO_3_ crystals (PDF # 01-083-0950). Additionally, the crystalline structure of N_2_-intercalated WO_3_ samples can be significantly affected by the addition of N_2_H_4_. Especially, the intensity of the (002) peak is higher than those of the other neighbor peaks of (020) and (200) for N_2_-intercalated WO_3_ samples, which is different from that of WO_3_-0 sample. This could suggest anisotropic progress of crystallization involving the predominant crystallization of (002) from n_W_:n_N2H4_ ratio of 1:0.32 to 1:2.5 followed by progressive crystallization of (020) and (200) with the ratio increased to 1:7.5, as depicted by calculation of the crystallite diameter of (002). The larger crystallite diameter of N_2_-intercalated WO_3_ samples (27–31 nm) than that of WO_3_-0 (17 nm) can be clearly observed, as shown in Table 1. The PEC performance of WO_3_ photoanodes for water splitting was extremely enhanced with highly uniform alignment along the (002) facet [48]. In the Raman spectra, the WO_3_ samples exhibited the characteristic peaks of the monoclinic WO_3_ at 134.4 cm^−1^ (lattice vibration), 270.6 cm^−1^ (*δ* (O-W-O) deformation vibration), 713.2cm^−1^, and 807.1 cm^−1^ (*δ* (O-W-O) stretching vibration) in the range of Raman shift from 100~1000 cm^−1^. Raman analysis also showed the tendency of Raman peaks to sharpen as the n_W_:n_(NH4)2S_ ratio increased, which is well consistent with the results of XRD analysis.

As shown in Figure 4, it could be seen that the lattice parameters a, b, c and *β* were remarkably influenced by the addition of N_2_H_4_ for the N_2_-intercalated WO_3_ samples (a = 7.3121(2)–7.3878 (3) Å, b = 7.4786(4)–7.5348 (2) Å, c = 7.6411 (2)–7.6948 (4) Å, *β* = 90.70(3)–90.78(2)). The values of parameters b, c, and *β* for all of samples decreased with the addition of N_2_H_4_ increase up to 1:2.5, and then increased at higher molar ratio of n_W_:n_N2H4_. However, the value of parameters an increased below 1:2.5 and then decreased with molar ratio of n_W_:n_N2H4_ increased. The lattice volumes of N_2_-intercalated WO_3_ samples were larger than that of WO_3_−0, and the largest lattice volume of 216.54(1) Å^3^ for WO_3_−2.5 was observed due to the insertion of highest contents for N_2_ into the lattice. Therefore, it can be confirmed that the N_2_ was intercalated into WO_3_ lattice by analyzing the change of lattice parameters and volume.

Raman spectra at high wavenumber (Figure 5) were investigated to clarify the configuration and existence of N_2_ into the WO_3_ lattice, which is responsible for the N content. No peak was observed for WO_3_-0, however, Raman spectra of N_2_-intercalated WO_3_ samples exhibited signals at 2327−2342 cm^−1^, which can be ascribed to the N≡N vibration of N_2_ in the WO_3_ lattice on the basis of N≡N vibration of gaseous N_2_ (2330 cm^−1^) and the N_2_-intercalated WO_3_ reported previously as well as relevant compounds [49,50,51,52]. Additionally, only one peak at 2330 cm^−1^ was observed for WO_3_−0.62. It can be attribute to the lower amount of N_2_ into the WO_3_−0.62 lattice, because such one peak was observed for N_2_-intercalted WO_3_ prepared by dodecylamine according to ours earlier report [30]. However, it is amazing to note that two peaks at 2328 cm^−1^ and 2342 cm^−1^ began to be observed from n_W_:n_N2H4_ of 1:1.2 as the content of N_2_ increased. This significant difference confirms that the addition of N_2_H_4_ affect not only the N content but also the configuration of N_2_ into the WO_3_ lattice. Further, The N_2_ intercalation can be also proved by our previously reported [17,30], the XPS spectrum in an W 4f region was deconvoluted by four bands at 37.0, 34.9, 32.9, 31.2 eV for N_2_ intercalated WO_3_ The bands at 37.0 and 34.9 eV in higher energy for N_2_ intercalated WO_3_ (37.1 and 34.9 eV for NH_3_-WO_3_) are assigned to 4f_5/2_ and W 4f_7/2_ of the WO_3_ lattice similarly to WO_3_−0. The bands at 32.9 and 31.2 eV in lower energy for N_2_ intercalated WO_3_ can be assigned to the binding energies of 4f_5/2_ and W 4f_7/2_ interacted with N_2_ intercalated. XPS data of WO_3_-N_2_H_4_, in which a peak at 396.9 eV assigned to the intercalated nitrogen bound with the tungsten center. This is obviously different from the band at 396 eV of N-doped metal oxides (WO_3_, TiO_2_, etc.) [53,54] and metal nitrides (TiN) [55]. Additionally, the nitrogen intercalation hardly causes the oxygen defects, which can be beneficial to improving the optical properties of WO_3_.

### 2.2. The Optical Properties of N_2_ Intercalated WO_3_

The optical properties of WO_3_ samples were investigated by UV-vis DRS (Figure 6A) and the corresponding Tauc plots (Figure 6B) of the WO_3_−0 (a) and N_2_ intercalated WO_3_ (b–f) samples with changes in the ratio of n_H2WO4_:n_N2H4_. Compared to the WO_3_−0 (469 nm), only a slight redshift of 11 nm in the absorption edge was observed for the WO_3_−0.62 (480 nm). However, a significant redshift can be seen in the absorption edge with new shoulders appeared in a longer wavelength region than WO_3_−0 with increased the ratio of n_H2WO4:_n_N2H4_. It is obvious that the absorption edges extended to the longer wavelength as the ratio of n_H2WO4_:n_N2H4_ was increased below 1:2.5, and then they decreased when further increased the addition of N_2_H_4_. Furthermore, the formation of the absorption shoulders exhibited the same trend as the formation of the peak at 2342 cm^−1^ in the Raman spectra, implying that the formation of this peak is beneficial to the generation of the absorption shoulders. According to the Tauc plots, the bandgap of WO_3_−0.62 (2.58 eV) was slightly reduced by 0.06 eV than WO_3_−0 (2.64 eV) due to the formation of a new intermediate N 2p orbital between the CB and the VB after intercalation of N_2_ into the WO_3_ lattice, as indicated by the earlier report [17,27,30]. As the ratio of n_H2WO4_:n_N2H4_ was increased from 1:1.2, the Tauc plots exhibited two different slopes due to the existence of the new shoulders, with the absorption energies derived from the slopes, as shown in Table 2.

Figure 7 illustrates the relation between the KM values at 500 nm (KM_500_). It was observed that the KM_500_ value is a measure of the increase/decrease of the shoulders at 470–600 nm. Compared with both WO_3_−0 and WO_3_-0.62, the KM_500_ increased from 0.11 to 0.31 with an increase in the ratio of n_H2WO4_:n_N2H4_ from 1:1.2 to 1:2.5, and thereafter, decreased from 1:2.5 to 0.13 at 1:7.5. The dependency of KM_500_ on the n_H2WO4_:n_N2H4_ ratios applies to the case of the N content, suggesting that the longer wavelength absorption due to the shoulders can be attributed to the intercalation of N_2_ into a WO_3_ lattice.

Mott–Schottky plots from alternating-current impedance measurements were taken to reveal the band structure of the N_2_ intercalated WO_3_ electrodes. As shown in Figure 8, typical behavior for n-type semiconductors was confirmed for all WO_3_ samples, because the reciprocal of the square of capacitance (C^−2^ [F^−2^ cm^4^]) linearly increased with the applied potentials beyond the flat band (*E*_FB_) potentials. The *E*_FB_ and the donor carrier densities (*N*_D_ [cm^−3^]) were provided from the *x*-intercept and the slopes of the straight line, respectively (Table 2). *E*_FB_ value of the WO_3_-0 electrode was 0.38 V, which was closed to those of earlier-reported WO_3_ electrodes (0.36–0.41 V vs. Ag/AgCl) [32,33]. However, the *E*_FB_ values of the N_2_ intercalated WO_3_ electrodes (0.23–0.36 V) were lower than that of WO_3_-0 electrode. The *N*_D_ values of the N_2_ intercalated WO_3_ electrodes were higher than that for WO_3_-0 electrode. Especially, the highest *N*_D_ value for WO_3_−2.5 (4.15 × 10^19^ cm^−3^) was calculated, which was 1.12, 1.09, 1.08, 1.03, and 1.06 times higher than those of WO_3_−0, WO_3_−0.62, WO_3_−1.2, WO_3_−5, and WO_3_−7.5 electrodes. The negative shift of the *E*_FB_ potential and the increase of the *N*_D_ are usually beneficial to the improvement of the PEC performance for water oxidation.

The band structures of the electrodes were estimated, and their energies are summarized in Table 2. As suggested by the previous report, for the N_2_ intercalated WO_3_, a new intermediate N 2p orbital could be formed between the CB and the VB of WO_3_ due to the intercalation of N_2_. The lower energies for the WO_3_−1.2, WO_3_−2.5, WO_3_−5, and WO_3_−7.5 were 2.17, 1.92, 2.01, and 2.08 eV, respectively, which can be attributed to excitation from the intermediate N 2p orbital to the CB of WO_3_. the different amount of N_2_ intercalated could be possible as an explanation of these difference in the lower energy values. The lower value for WO_3_−2.5 is caused by the high amount of N_2_ intercalated. The potentials of the intermediate bands (*E*_IB_) were calculated from the *E*_FB_ and the excitation energies to be 2.51, 2.15, 2.31, and 2.41 V vs. Ag/AgCl for WO_3_−1.2, WO_3_−2.5, WO_3_−5, and WO_3_−7.5, respectively. The higher energies (2.58, 2.55, 2.45, 2.52, and 2.51 eV) for WO_3_−0.62, WO_3_−1.2, WO_3_−2.5, WO_3_−5, and WO_3_−7.5 can be ascribed to the main band gap excitation. The VB potentials (E_VB_) are calculated to be 2.94, 2.89, 2.68, 2.82, and 2.83 V vs. Ag/AgCl for WO_3_-0.62, WO_3_-1.2, WO_3_-2.5, WO_3_-5, and WO_3_-7.5, respectively.

### 2.3. Photoelectrochemical Properties

The LSVs for these electrodes were measured with chopped visible light irradiation to study their PEC water oxidation performance. The onset potential for photocurrents was 0.2 V vs. Ag/AgCl on visible-light irradiation for these electrodes due to water oxidation (Figure 9A). The photocurrent of 1.08 mA cm^−2^ at 1.0 V for WO_3_−2.5 was the highest.

Figure 9B exhibits that the photocurrent at 0.68 V vs. Ag/AgCl under visible-light irradiation chopped was stable during PEC water oxidation (5 min) for all of electrodes. The photocurrent of the WO_3_−2.5 electrode (0.85 mA cm^−2^) was higher than those of the WO_3_−0 (0.02 mA cm^−2^), WO_3_−0.62 (0.19 mA cm^−2^), WO_3_−1.2 (0.43 mA cm^−2^), WO_3_−5 (0.63 mA cm^−2^), and WO_3_−7.5 (0.58 mA cm^−2^) by a factor of 42.5, 4.5, 2.0, 1.3, and 1.5, respectively.

Figure 10 shows the action spectra of IPCE at 0.5 V vs. Ag/AgCl for different electrodes. The photocurrent could only be observed below 470 nm for WO_3_−0, which is consistent with the bandgap energy of WO_3_ [1]. For the WO_3_−0.62 electrode, the onset wavelength for photocurrent generation was at 480 nm (2.58 eV) after N_2_ intercalation. The onset wavelengths for WO_3_−1.2, WO_3_−2.5, WO_3_−5, and WO_3_−7.5 are considerably shifted to the wavelengths (530 nm) longer than that of WO_3_−0.62. Moreover, the IPCE results suggest that the photocurrent was generated based on the bandgap excitation, and the bandgap excitation occurs through collateral excitation from intermediate N 2p orbital to CB for the N_2_ intercalated WO_3_ electrodes. However, for all N_2_ intercalated electrodes, the photocurrent at wavelengths longer than 530 nm could not be detected due to the limited current detection level of the employed apparatus.

Photoelectrocatalysis was conducted under the visible light irradiation (*λ* > 450 nm, 100 mW cm^−2^) at potentiostatic conditions of 0.5 V vs. Ag/AgCl (1.05 V vs. RHE) in a 0.1 M phosphate buffer (pH 6.0) for 1 h using different electrodes (Figure 11). A higher photoanodic current due to water oxidation was observed for the WO_3_−2.5 electrode. The highest charge amount passed and the amount (n_O2_) of O_2_ evolved during the 1 h photoelectrocatalysis for WO_3_−2.5 was 2.05 C and 5.19 mmol (97% Faradaic efficiency), respectively, compared with the electrodes prepared at other conditions (Table 3). These results clearly prove that the intercalation of N_2_ enhances the PEC performance of WO_3_−2.5 in application to water oxidation. The decay of photoanodic currents was observed for all electrodes. This can be attributed to the formation of inactive tungsten-peroxo adducts [15] and adherence of O_2_ bubbles [56].

The Tafel plots and the measurement of electrochemical impedance are useful to evaluate the electron transport and its influence on PEC performance for water oxidation. In the Figure 12A, the Tafel plots of WO_3_−2.5 for water oxidation exhibited the lowest Tafel slope (d, 25 ± 0.3 mVdec^−1^) compared that of WO_3_-0 (a, 35 ± 0.2 mVdec^−1^), WO_3_−0.62 (b, 33 ± 0.4 mVdec^−1^), WO_3_−1.2 (c, 30 ± 0.3 mVdec^−1^), WO_3_−5 (e, 28 ± 0.5 mVdec^−1^)and WO_3_−7.5 (f, 29 ± 0.4 mVdec^−1^), suggesting the resistance of electrochemical reaction of WO_3_−2.5 electrodes is less than those of the other WO_3_ electrodes. As shown in the Figure 12B, only one semicircle was observed for all WO_3_ electrodes in Nyquist plots, indicating that charge transfer process plays an important role in the PEC reaction. WO_3_−2.5 gave smaller semicircles than those of WO_3_ electrodes, which implies that the WO_3_−2.5 electrode has lower charge transfer resistance and higher separation efficiency of photogenerated electron-hole pairs than others electrodes. As the ratio of n_H2WO4_:n_N2H4_ increased from 1:0 to 1:2.5, the semicircles decreased and then increased from 1:5 to 1:7.5, corresponding to the amounts of N_2_ into the WO_3_ lattice.

## 3. Materials and Methods

### 3.1. Materials

Tungstic acid (H_2_WO_4_), hydrazine monohydrate (N_2_H_4_·H_2_O), Marpolose (60MP-50), and Polyethylene glycol (PEG, Mw = 2000) were obtained from McLean’s Reagent (Shanghai Macklin Biochemical Co.,Ltd., shanghai, China). A Fluorine doped tin oxide (FTO)-coated glass substrate was obtained from Dalian HeptaChroma Co. Ltd. (Dalian, China), Millipore water (DIRECT-Q 3UV, Merck Ltd. Merck Ltd., Shanghai, China) was used to prepare the solutions. All of the chemicals were of analytical grade and were used as received unless mentioned otherwise.

### 3.2. Synthesis of N_2_-Intercalated WO_3_

According to the approach we reported previously [17], N_2_H_4_·H_2_O (36.5–438 μL, 0.75–9.0 mmol) solution were added to H_2_WO_4_ (0.3 g, 1.2 mmol) under vigorous stirring at room temperature to form a yellow suspension with H_2_WO_4_:N_2_H_4_ molar ratio (n_W:_n_N2H4_) of 1:0.62–7.5 in 3 mL water. The white N_2_H_4_-derived precursor was obtained after the suspension was slowly evaporated. The N_2_H_4_-derived precursor powders were heated at 420 °C (1 °C min^−1^) for 1.5 h in flowing O_2_ to obtain different WO_3_ samples. A control WO_3_ sample was prepared in the same manner without addition of N_2_H_4_·H_2_O.

### 3.3. Fabrication of Electrodes

In a typical procedure, 0.6 mL of water was added to the N_2_H_4_-derived precursor powder (0.8 g), PEG (0.2 mg), and Marpolose (80.0 mg), and it was slowly stirred until a smooth paste without bubbles was formed. A doctor-blading method was employed to coat the resulting paste over a clean FTO glass substrate (1.0 cm^−2^ area) and dried at 80 °C for 15 min. The N_2_H_4_-derived precursor electrodes were calcined at 420 °C in flowing O_2_ flow to give different WO_3_ electrodes, which are denoted as WO_3_-0.62, WO_3_-1.2, WO_3_-2.5, WO_3_-5 and WO_3_-7.5, respectively. The control WO_3_ electrode was fabricated by the same method using a precursor prepared without addition of N_2_H_4_·H_2_O, denoted as WO_3_-0.

### 3.4. Measurement

Characterization of the crystalline phase was performed by powder X-ray diffraction (XRD) using a monochromated (Shimadzu International Trade (Shanghai) Co., Ltd., Shanghai, China, XRD-6000, Cu Kα λ = 1.54 Å). The electron microscopy images of surface morphology were observed using a field emission scanning electron microscope (JEOL, TSM-6510LV, Japan). The energy-dispersive X-ray spectroscopic (EDS) data were collected using an electron probe microanalysis (JED-2300, JEOL, Tokyo, Japan) operated at an accelerating voltage of 10 kV. Raman spectra were collected using a Raman microspectroscopic apparatus (Horiba-Jobin-Yvon LabRAM HR, Paris, France) using 532 nm excitation and silicon standard wavenumber (520.7 cm^−1^). A Thermo Fisher Scientific (Thermo Fisher Scientific (China) Co., Ltd., Shanghai, China) ESCALAB Xi+ instrument was employed to collect the XPS spectra, and calibrated by the C 1s peak, appearing at 284.2 eV. A spectrophotometer (Shimadzu UV-2700) in a DR mode with an integrating sphere (ISN-723) was taken to record the UV-visible DRS were recorded.

PEC measurements were examined using an electrochemical analyzer (Shanghai Chenhua Instrument Co., Ltd., CHI760E). A two-compartment PEC cell separated by a Nafion membrane. A three-electrode system has been employed using a WO_3_ electrode and Ag/AgCl electrode as the working and reference electrodes in one compartment, and a Pt wire in the other compartment as the counter electrode. All the PEC experiments were taken in an aqueous 0.1 M phosphate buffer solution (pH 6.0). The linear sweep voltammograms (LSV) were measured at a scan rate of 5 mV s^−1^ between −0.2 V and 1.0 V. Light (λ > 450 nm, 100 mW cm^−2^) was irradiated from the backside of the working electrode using a 500W xenon lamp with a UV-cut filter (*λ* > 450 nm) and liquid filter (0.2 M CuSO_4_, 5.0 cm light pass length) for cutting of heat ray. Electrochemical impedance spectra were measured at an applied potential of 0.68 V vs. Ag/AgCl (1.23 V vs. RHE) in a frequency range from 10 mHz to 20 kHz (amplitude of 50 mV). The output of light intensity was calibrated as 100 mW cm^−2^ using a spectroradiometer (Ushio Inc., USR-40, Ushio Shanghai Inc., Shanghai, China). Photoelectrocatalysis was conducted under the potentiostatic conditions at 0.5 V at 25 °C with illumination of light (*λ* > 450 nm, 100 mW cm^−2^) for 1 h. The amounts of H_2_ and O_2_ evolved were determined from the analysis of the gas phase of counter and working electrode compartments, respectively, using gas chromatography (Shimadzu GC-8A with a TCD detector and molecular sieve 5A column and Ar carrier gas). A monochromic light with 10 nm of bandwidth was employed using a 500 W xenon lamp with a monochromator for IPCE measurements.

## 4. Conclusions

N_2_ intercalated WO_3_ was controllably synthesized using N_2_H_4_ with a dual functional role, namely as an N atom source for N_2_ intercalation and as a structure-directing agent for the nanorod architecture. The addition of N_2_H_4_ dependence on the physiochemical properties and the performance of the PEC water oxidation of the WO_3_-0 and N_2_ intercalated WO_3_ electrodes were investigated to characterize N_2_ intercalation into the WO_3_ lattice and reveal the mechanism of the superior performance of PEC water oxidation for the N_2_ intercalated WO_3_ photoanode. The N_2_ intercalated WO_3_ exhibited the optimum n_W_:n_(NH4)2S_ ratio at 1:2.5 for the high concentration of N elements. The N_2_ intercalation is responsible for the significant red shift in the absorption edge, with a new shoulder appearing at 470–600 nm compared to that of WO_3_-0. The N_2_ intercalated WO_3_ photoanode is able to utilize visible light in longer wavelengths below 530 nm for PEC water oxidation, in contrast to utilization below 470 nm for the WO_3_-0 photoanode. The N_2_ intercalated WO_3_ photoanode is expected to be applied for PEC water splitting cells in artificial photosynthesis to improve the solar energy conversion efficiency.

## Figures and Tables

**Figure 1 molecules-28-02987-f001:**
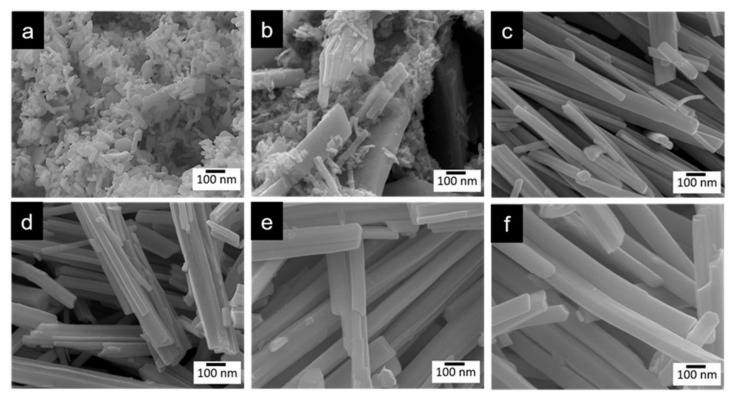
SEM images of (**a**) WO_3_−0, (**b**) WO_3_−0.62, (**c**) WO_3_−1.2, (**d**) WO_3_−2.5, (**e**) WO_3_−5, and (**f**) WO_3_−7.5.

**Figure 2 molecules-28-02987-f002:**
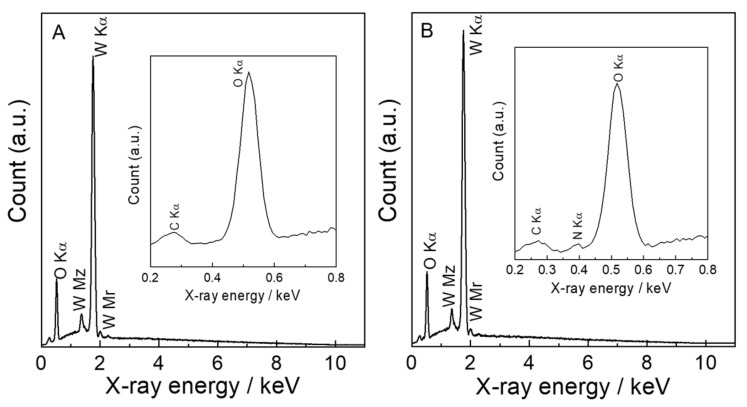
The EDS data of (**A**) WO_3_−0 and (**B**) WO_3_−2.5 in the range of 0–11 keV. (Insert) The magnified EDS data of WO_3_−0 and WO_3_−2.5 in the range of 0.2–0.8 keV, respectively.

**Figure 3 molecules-28-02987-f003:**
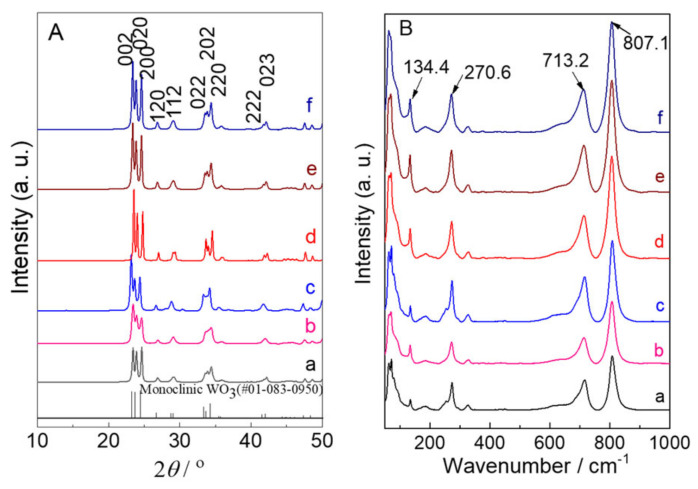
(**A**) XRD patterns and (**B**) Raman spectra of (a) WO_3_−0, (b) WO_3_−0.62, (c) WO_3_−1.2, (d) WO_3_−2.5, (e) WO_3_−5, and (f) WO_3_−7.5, respectively.

**Figure 4 molecules-28-02987-f004:**
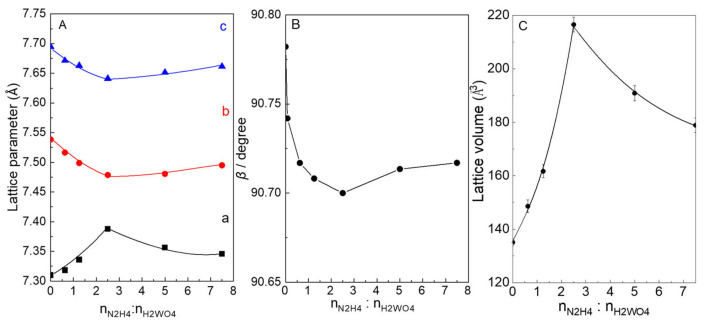
(**A**) Plots of lattice parameters of (a), (b), and (c) versus the addition of N_2_H_4_. (**B**) Plots of *β* versus the addition of N_2_H_4_, and (**C**) Lattice volumes of N_2_-intercalated WO_3_ samples prepared with addition of N_2_H_4_. The lattice volume (V) was calculated according to equation: V = *abc*(sin*β*), *β* is the angle between a and c.

**Figure 5 molecules-28-02987-f005:**
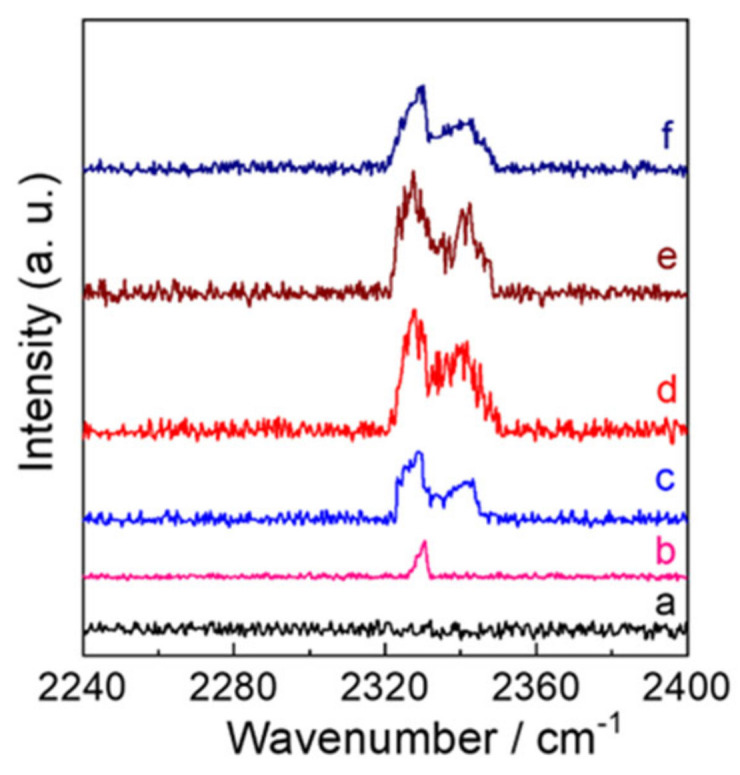
Raman spectra in the wavenumber region of 2240–2400 cm^−1^ for (a) WO_3_−0, (b) WO_3_−0.62, (c) WO_3_−1.2, (d) WO_3_−2.5, (e) WO_3_−5, and (f) WO_3_−7.5, respectively.

**Figure 6 molecules-28-02987-f006:**
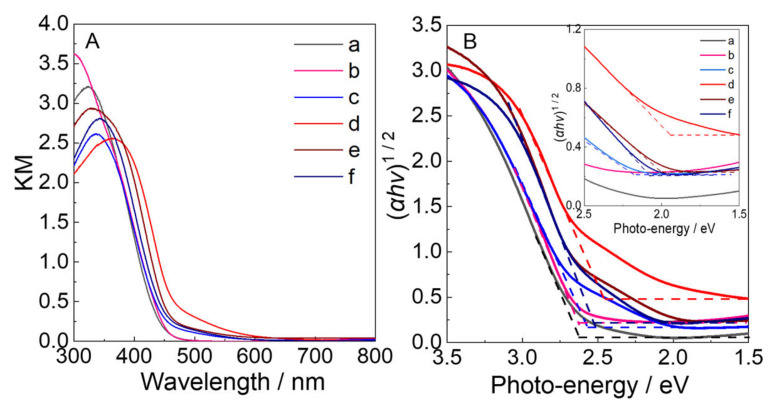
(**A**) UV-Visible DRS and (**B**) Tauc plots based on (a) WO_3_−0, (b) WO_3_−0.62, (c) WO_3_−1.2, (d) WO_3_−2.5, (e) WO_3_−5, and (f) WO_3_−7.5. Insets show the magnified spectra near the edges.

**Figure 7 molecules-28-02987-f007:**
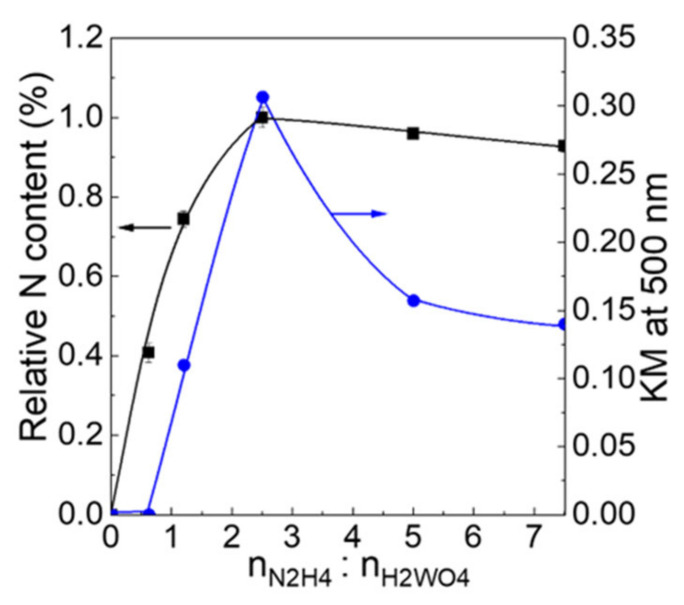
Relationship between the relative contents of N and n_H2WO4_:n_N2H4_ ratio. The relative N contents were measured in EDS data and normalized by the highest contents for n_H2WO4_:n_N2H4_ of 1:2.5.

**Figure 8 molecules-28-02987-f008:**
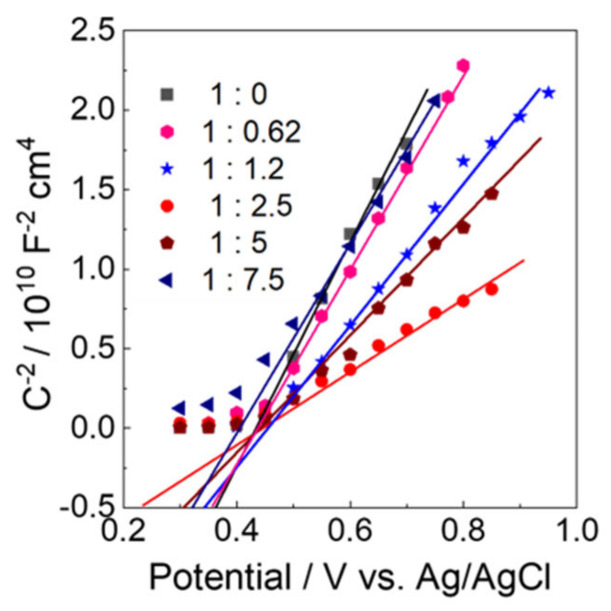
Mott-Schottky plots of (black) WO_3_−0, (pink) WO_3_−0.62, (blue) WO_3_−1.2, (red) WO_3_−2.5, (wine) WO_3_−5, and (navy) WO_3_−7.5 electrodes in a 0.1 M phosphate buffer solution of pH 6.0. frequency, 0.1 Hz; amplitude potential, 10 mV.

**Figure 9 molecules-28-02987-f009:**
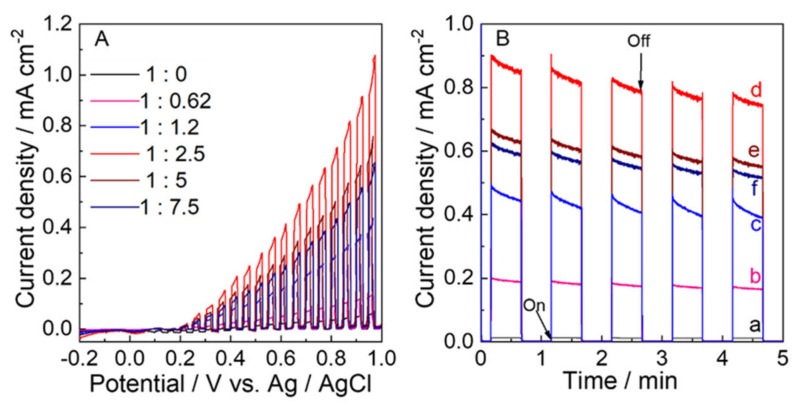
(**A**) Linear sweep voltammograms (LSV), and (**B**) Time course of the photocurrent at 0.68 V vs. Ag/AgCl (1.23 V vs. RHE) of the (a) WO_3_−0, (b) WO_3_−0.62, (c) WO_3_−1.2, (d) WO_3_−2.5, (e) WO_3_−5, and (f) WO_3_−7.5 electrodes with visible-light irradiation chopped in a 0.1 M phosphate buffer solution of pH 6.0 with visible-light irradiation (*λ* > 450 nm, 100 mW cm^−2^).

**Figure 10 molecules-28-02987-f010:**
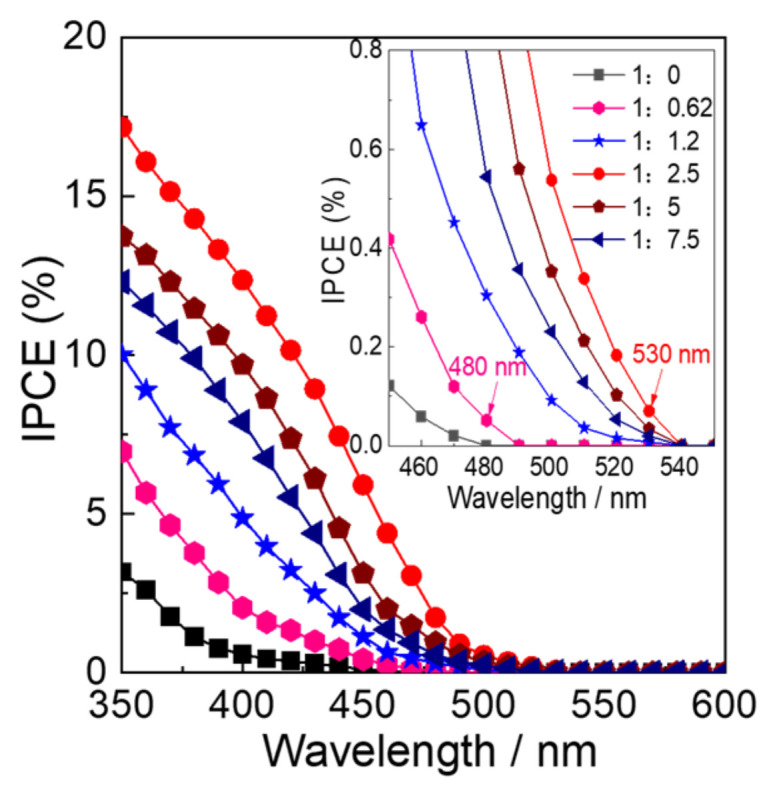
Action spectra of IPCE of the (black) WO_3_−0, (pink) WO_3_−0.62, (blue) WO_3_−1.2, (red) WO_3_−2.5, (wine) WO_3_−5, and (navy) WO_3_−7.5 electrodes.

**Figure 11 molecules-28-02987-f011:**
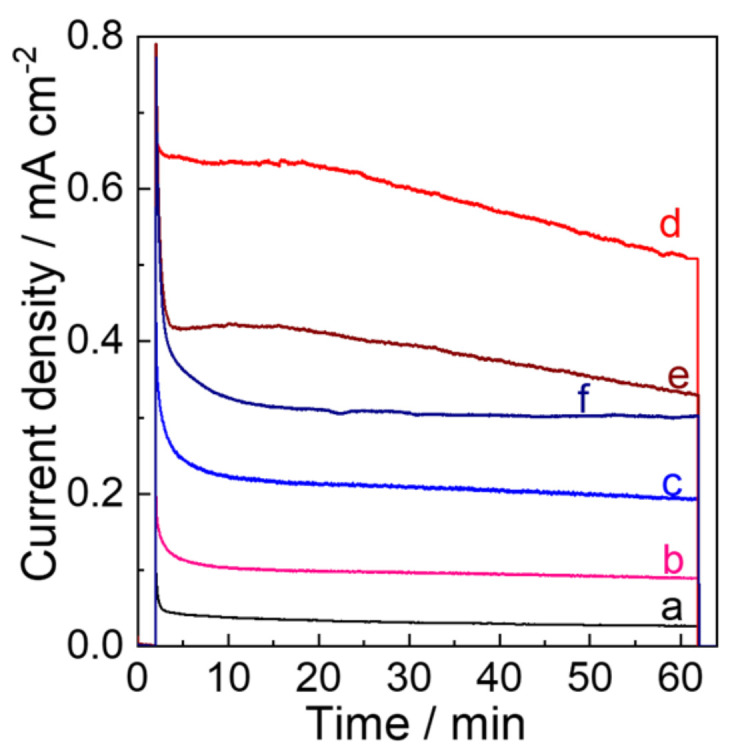
Photocurrent density versus time profiles during PEC water oxidation in a 0.1m phosphate buffer solution of pH 6.0 at 0.68 V vs. Ag/AgCl (1.23 V vs. RHE) upon visible-light irradiation (*λ* > 450 nm, 100 mWcm^−2^) using (a) WO_3_−0, (b) WO_3_−0.62, (c) WO_3_−1.2, (d) WO_3_−2.5, (e) WO_3_−5, and (f) WO_3_−7.5 electrodes.

**Figure 12 molecules-28-02987-f012:**
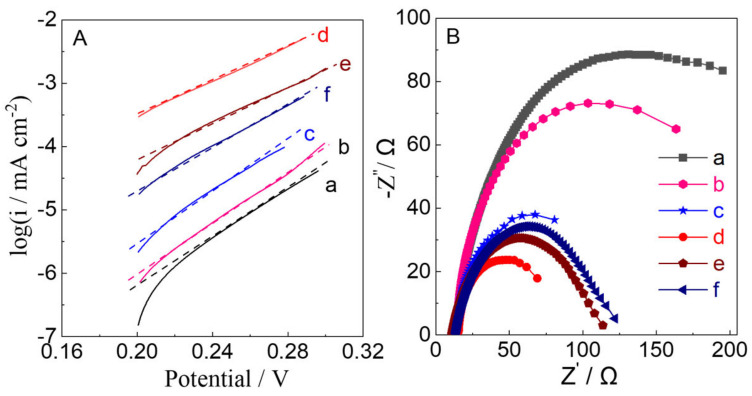
(**A**) Tafel plots and (**B**) Nyquist plots of (a) WO_3_−0, (b) WO_3_−0.62, (c) WO_3_−1.2, (d) WO_3_−2.5, (e) WO_3_−5, and (f) WO_3_−7.5 electrodes for photoelectrocatalytic water oxidation in a 0.1 M phosphate buffer solution (pH = 6).

**Table 1 molecules-28-02987-t001:** Summary of physicochemical properties of different WO_3_ samples.

Samples	Molar Ratio of W:N ^a^	Crystallite Diameter ^b^ (nm)	Surface Area ^c^ (m^2^ g^−1^)
WO_3_−0	1:0	17	9.6
WO_3_−0.62	1:0.040	22	12.1
WO_3_−1.2	1:0.073	25	16.6
WO_3_−2.5	1:0.098	31	21.2
WO_3_−5	1:0.096	30	20.4
WO_3_−7.5	1:0.093	27	17.3

^a^ The local content of N was analysed according to the approach we reported previously [17,30]. ^b^ The crystallite diameters were calculated from XRD data according to Scherrer equation. ^c^ The surface areas were provided form N_2_ sorption isotherms.

**Table 2 molecules-28-02987-t002:** Summary of optical and electrochemical properties and energies of band structures of various WO_3_ samples.

Samples	Absorption Energies	*E* _FB_	*N*_D_ (10^19^ cm^−3^)	*E* _IB_	*E* _VB_
WO_3_−0	2.64, -	0.38	3.68	-	3.02
WO_3_−0.62	2.58, -	0.36	3.78	-	2.94
WO_3_−1.2	2.55, 2.17	0.34	3.82	2.51	2.89
WO_3_−2.5	2.45, 1.92	0.23	4.15	2.15	2.68
WO_3_−5	2.52, 2.01	0.30	4.01	2.31	2.82
WO_3_−7.5	2.51, 2.08	0.32	3.91	2.41	2.83

**Table 3 molecules-28-02987-t003:** Summary of PEC water oxidation in a 0.1 M phosphate buffer solution (pH 6.0) for 1 h using different WO_3_ electrodes calcined at 420 °C.

Samples	Charge /C	*n*_O2_ /μmol	F.E._O2_ ^a^ (%)	*n*_H2_^b^ /μmol	F.E._H2_ ^c^ (%)
WO_3_−0	0.08	0.11	54	0.34	83
WO_3_−0.62	0.32	0.75	91	1.56	95
WO_3_−1.2	0.71	1.72	92	3.53	96
WO_3_−2.5	2.05	5.19	97	10.6	100
WO_3_−5	1.29	3.16	94	6.58	98
WO_3_−7.5	1.18	2.83	92	5.98	98

^a^ Faradic efficiency of O_2_ evolution. ^b^ *n*_H2_ is the amount of H_2_ evolved in the Pt counter electrode compartment. ^c^ Faradic efficiency of H_2_ evolution.

## Data Availability

Not applicable.

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
