# Peer review of "Controllable Synthesis of N2-Intercalated WO3 Nanorod Photoanode Harvesting a Wide Range of Visible Light for Photoelectrochemical Water Oxidation"

_molecules, 2023, doi:10.3390/molecules28072987_

Round 1
Reviewer 1 Report
See the attachment for details.

Author Response
Response to Reviewer 1 Comments
Thank you very much for your kind and helpful comments. We revised the manuscript carefully considering your comments.
Point 1: The author proposes that the introduction of N2H4 has a dual function, one of which is to be used as a nano-structure guiding agent. How to prove it? I wonder if the author considers using other materials instead of N2H4 to provide N source for WO3. What will the result be?
Response 1: We have synthesized the N2 intercalated WO3 using NH3·H2O (ChemSusChem 2018, 11, 1151-1156; ACS Sustainable Chem. Eng., 2019, 7, 17896-17906.), in which the massive blocks of N2 intercalated WO3 was observed in the microscale SEM image. Additionally, we also reported the WO3 particles utilizing the dodecylamine (DDA) as a surfactant template (Chem. Eur. J. 2017, 23, 6596–6604.). Recently, we prepared the S and N co-doping WO3 with blocklike particles morphology using ammonium sulfide ((NH4)2S) as N source (Nanomaterials 2022, 12, 2079–2091.). These results indicate that the using of N2H4 is beneficial for guiding the formation of nanorods.
Point 2: In the last paragraph of the introduction, it is pointed out that the relationship between calcination temperature and photocatalysis performance is studied, but only one temperature is used in this manuscript. Has the author considered the effect of different calcination temperature on the crystallinity of N2‑Intercalated WO3 Nanorod?
Response 2: As you pointed out, the calcination temperature can influence the crystallinity of N2‑Intercalated WO3 Nanorod dramatically. We have already investigated the the calcination temperature dependence on the crystallinity and photoelectrochemical performance for water oxidation of N2‑Intercalated WO3 Nanorod previously (ACS Sustainable Chem. Eng., 2019, 7, 17896-17906). There, N2‑Intercalated WO3 Nanorod calcined 420oC exhibited the higher N contents than other temperatures. Thus, we selected this temperature in this issue. Additionally, we considered that the addition of N2H4 could influence the contents of N2 into the WO3 lattice, as well as the morphology of the N2-intercalated WO3, and also the PEC performance for water oxidation. Therefore, it is necessary to further research the dependence of the addition of N2H4 on the physiochemical properties and the PEC performance, because it have not been investigated yet.
Point 3: Figure 2 (A) shows that the obvious left shift of the strongest peak of WO3-1.25 represents lattice expansion, while the right shift of the peak at WO3-2.5 represents lattice contraction. What is the reason for this?
Response 3: As you point out, the lattice was expanded with increasing the ratio of nN2H4:nH2WO4 from 1:0.62 to 1:1.2 due to a significant increase of the amount of intercalated N2, compared with WO3-0. And thereafter, the lattice contraction was observed at higher ration due to the capacity of the WO3 lattice for N2 intercalation, which was confirmed from the results of lattice parameters of a, b, and c versus the addition of N2H4 (Figure 4).
Point 4: Figure 6 shows that the relative content of N decreases with the increase of the ratio of nN2H4:nH2WO4 after WO3-2.5. Why does the relative content of N decrease with the increase of N2H4?
Response 4: The decrease of the relative content of N with the increase of N2H4 can be corresponding to the limitations in the intercalation capacity of the WO3 lattice. This result consists with the trend of lattice volume of WO3 after N2 intercalation.
Point 5: Introduce different ratios of nN2H4:nH2WO4, and oxidize the white derivative precursor powder of N2H4 at 420 ℃ for 1.5h. How does the author judge that N2H4 has been completely reacted?
Response 5: According to the TG and XRD results we reported previously, the N2H4 can be completely reacted. (ChemSusChem 2018, 11, pp 1151-1156; ACS Sustainable Chem. Eng., 2019, 7, 17896-17906.).
Figure 1 (A) XRD patterns of (a) N2H4 derived precursor and (B) (b) N2H4-WO3 calcined at 420oC. (B) TG data of N2H4 derived precursor.
Point 6: Some views should be supported by references, such as “According to the approach we reported previously…”
Response 6: It has been added in the revised manuscript.
Point 7: The figures and text in the draft need to be further optimized and professionalized to facilitate understanding and eliminate misunderstandings.
Response 7: It has been mentioned correctly in the revised manuscript.
We hope that the revision meets your comments and the revised version is accepted for publication.
Reviewer 2 Report
In this work, Li and coworkers intercalated N atoms into WO3 by adding N2H4 during the preparation process. It modified the morphology, crystallinity and light absorption properties and PEC activities of the WO3. The authors provided detailed characterization and discussion about the N-WO3 products. This work gives a good experimental evidence for the enhancing effect of N element on WO3 in PEC reaction. Therefore, it is recommended for publish of this paper in Journal of Molecules. Some comments are listed below.
1. The morphology of WO3 was changed by adding N2H4 in the precursor. From SEM and XRD, it shows that the exposed facets of WO3 are changed to some extent. It is suggested to have some comments on the effect of facet issue on the PEC activity, such as a recent work in Chemistry - A European Journal, 2022 (doi.org/10.1002/chem.202201169).
2. In Table 3, the Faradic efficiency for H2 evolution on Pt is also less than unit and it shows the same trend with that of O2 evolution on N-WO3. This result is interesting. What causes the low efficiency for H2 evolution?
Author Response
Response to Reviewer 2 Comments
Thank you very much for your kind and helpful comments. We revised the manuscript carefully considering your comments.
Point 1: The morphology of WO3 was changed by adding N2H4 in the precursor. From SEM and XRD, it shows that the exposed facets of WO3 are changed to some extent. It is suggested to have some comments on the effect of facet issue on the PEC activity, such as a recent work in Chemistry - A European Journal, 2022 (doi.org/10.1002/chem.202201169).
Response 1: As you pointed out, we cite the reference (Chem. Eur.J., 2022, 28, pp: e202201169- e202201177) to illustrate the effect of the (002) facet on the improvement of PEC water oxidation performance.
Point 2: In Table 3, the Faradic efficiency for H2 evolution on Pt is also less than unit and it shows the same trend with that of O2 evolution on N-WO3. This result is interesting. What causes the low efficiency for H2 evolution?
Response 2: The low efficiency for H2 evolution is explained by gradual photo-oxidation of the WO3 surface by holes to form tungsten-peroxo adducts (Chem. Mater. 2011, 23, 1105-1112.). With the ability of water oxidation decreased, at the meanwhile, the ability of water reduction also decreased. And thus, the efficiency for H2 evolution decreased.
We hope that the revision meets your comments and the revised version is accepted for publication.
Reviewer 3 Report
This manuscript studied the effect of the ratio between N2H4 and H2WO4 precursor on the synthesized N2-intercalated WO3 nanorod photoanode. Based on a series of structural, optical, and photoelectrochemical characterizations, the authors found the optimal precursor ratio. The work is well organized. I recommend the acceptance, though there are still some concerns that for the authors to address.
1. Please provide the reference of the “previously reported” work mentioned in the last paragraph of Introduction.
2. The graphic illustration of EDS results should be added in order to be more convincing than the numeric results.
3. The format of subfigure number differs in Figure 1 from other figures.
4. After N2 intercalation, is the oxidation state of W affected? I think it is important to perform XPS measurement, especially for the study that involves heterostructure with multiple compounds.
5. Figure 5B: what is F in the y-axis label? Is it alpha? Besides, the x-axis label of both main plot and inset should be “photon energy” rather than “photo energy”
6. Line 219: 0.42E20 -> 4.2E19
7. Some figures are too small so that details are not readable.
Author Response
Response to Reviewer 3 Comments
Thank you very much for your kind and helpful comments. We revised the manuscript carefully considering your comments.
Point 1: Please provide the reference of the “previously reported” work mentioned in the last paragraph of Introduction.
Response 1: As you pointed out, we cite the reference (ChemSusChem 2018, 11, 1151-1156) to illustrate the previous work.
Point 2: The graphic illustration of EDS results should be added in order to be more convincing than the numeric results.
Response 2: In order to illustrate the existence and the analytical method for the EDS data. The EDS figures of WO3-0 and WO3-2.5 were added in the revised manuscript.
Point 3: The format of subfigure number differs in Figure 1 from other figures.
Response 3: It has been mentioned correctly in the revised manuscript.
Point 4: After N2 intercalation, is the oxidation state of W affected? I think it is important to perform XPS measurement, especially for the study that involves heterostructure with multiple compounds.
Response 4: We have taken the measurement to investigate the effect of N2 intercalation on the oxidation state of W (ChemSusChem 2018, 11, 1151-1156), in which the XPS spectrum in an W 4f region was deconvoluted by four bands at 37.0, 34.9, 32.9, 31.2 eV for N2 intercalated WO3 The bands at 37.0 and 34.9 eV in higher energy for N2 intercalated WO3 (37.1 and 34.9 eV for NH3-WO3) are assigned to 4f5/2 and W 4f7/2 of the WO3 lattice similarly to WO3-0 (Mater. Lett. 2012, 82, 214-216.). The bands at 32.9 and 31.2 eV in lower energy for N2 intercalated WO3 can be assigned to the binding energies of 4f5/2 and W 4f7/2 interacted with N2 intercalated.
Point 5: Figure 5B: what is F in the y-axis label? Is it alpha? Besides, the x-axis label of both main plot and inset should be “photon energy” rather than “photo energy”
Response 5: As you pointed out, it has been correctly in the revised manuscript.
Point 6: Line 219: 0.42E20 -> 4.2E19
Response 6: It has been rewritten in the revised manuscript.
Point 7: Some figures are too small so that details are not readable.
Response 7: It has been modified in the revised manuscript.
Round 2
Reviewer 1 Report
The author answered all questions and made reasonable changes. I agree to accept the paper.